# Natural Sources of Selenium as Functional Food Products for Chemoprevention

**DOI:** 10.3390/foods12061247

**Published:** 2023-03-15

**Authors:** Małgorzata Dobrzyńska, Sławomira Drzymała-Czyż, Dagmara Woźniak, Sylwia Drzymała, Juliusz Przysławski

**Affiliations:** Department of Bromatology, Poznan University of Medical Science, Rokietnicka 3 Street, 60-806 Poznan, Poland

**Keywords:** selenium, selenium-rich product, functional food, Brazil nuts, *Brassica* species, cancer prevention

## Abstract

Cancer is one of the leading causes of death worldwide, the incidence of which is increasing annually. Interest has recently grown in the anti-cancer effect of functional foods rich in selenium (Se). Although clinical studies are inconclusive and anti-cancer mechanisms of Se are not fully understood, daily doses of 100–200 µg of Se may inhibit genetic damage and the development of cancer in humans. The anti-cancer effects of this trace element are associated with high doses of Se supplements. The beneficial anti-cancer properties of Se and the difficulty in meeting the daily requirements for this micronutrient in some populations make it worth considering the use of functional foods enriched in Se. This review evaluated studies on the anti-cancer activity of the most used functional products rich in Se on the European market.

## 1. Introduction

Cancer is one of the leading causes of mortality worldwide, accounting for nearly 10 million deaths in 2020. The most common cancer is female breast cancer (11.7%), followed by lung (11.4%), colorectal (10.0%), prostate (7.3%), and stomach (5.6%) cancers [1,2]. According to Global Cancer Statistics (GLOBOCAN 2020: Estimates of Incidence and Mortality Worldwide for 36 Cancers in 185 Countries), the number of people with cancer will continue to grow and reach 28.4 million by 2040 [3]. The main causes of the increase in cancer incidence are environmental and lifestyle factors, with an estimated 30–40% of all cancers preventable through lifestyle and diet [4,5,6].

In recent years, the chemopreventive effect of diet has focused on selenium (Se), and most epidemiological studies and clinical trials support the protective role of Se against cancer development. However, high doses of Se supplements must be consumed for an anti-cancer effect. The multi-centre, double-blind, randomised, placebo-controlled cancer prevention trial by Clark et al. reported that supplementation of 200 µg of Se per day can reduce the incidence of and mortality from carcinomas [7]. Nonetheless, some studies have shown an inverse association between Se exposure and risk of some cancer types. The randomised, prospective, double-blind SELECT (Selenium and Vitamin E Cancer Prevention Trial) study indicated that the supplementation (mean of about 5 years) of selenomethionine and vitamin E does not prevent prostate cancer in the generally healthy population, with a statistically non-significant increase in prostate cancer in the vitamin E-alone group and non-significant increase in diabetes mellitus associated with the Se-alone group. There are several possible explanations for why Se did not prevent prostate cancer in men in the SELECT study [8,9,10,11]. The supplements may have exceeded the dose at which the protective effect is observed, and the study participants may have had different levels of Se intake before the study. Se possibly reduces the risk of prostate cancer, but only in men with Se deficiency [12]. The Cochrane reviews on Se for cancer prevention have shown that those with higher Se levels have a lower incidence of cancer. However, the lower risk of cancer may be related to other factors that reduce the cancer risk, such as a healthier diet or lifestyle [13]. Therefore, an additional analysis of Se action in the context of cancer prevention is necessary [9,13].

Se consumption varies considerably throughout the world, with some diets being Se deficient and some having an excess of Se (Eastern European countries 10−30 µg Se/day vs. Venezuela 200−350 µg Se/day) [14,15]. Se action is important in the human body (e.g., structural and enzymic roles, and antioxidant activity catalyses the production of active thyroid hormone [16]), and thus Se deficiency is a major global problem [17]. According to nutritional requirements, the adequate intake (AI) of Se is 70 µg/day for adults, 15−65 µg/day for children depending on age, and 85 µg/day for lactating women [18]. Regarding cancer prevention, some researchers suggest a daily dose of 100–200 µg of Se to inhibit genetic damage and cancer development [19]. The Scientific Committee for Food (SCF) adopts the tolerable upper level (UL) of Se for adults as 255 µg/day [20]. Consequently, functional food rich in Se is of interest due to difficulties in meeting the daily Se requirements and its beneficial anti-cancer properties. Thus, this review evaluated studies on the anti-cancer activity of functional products rich in Se identified in PubMed, Scopus, Cochrane Library, Web of Science, and Embase using the following keywords: “selenium enriched” and “cancer” or “selenium rich” and “cancer” or “selenium enriched” and “chemoprevention” or “selenium rich” and “chemoprevention”. An additional search was performed for Brazil nuts (on account of having the highest Se content naturally occurring in food), *Brassica* species (as a source of Se-methylselenocysteine), *Allium* species (one of the most widely consumed vegetables worldwide), and Se-enriched yeast (evaluated in human clinical trials). English language articles published until 30 January 2023 were reviewed (Figure 1).

## 2. Dietary Sources of Selenium Compounds and Their Metabolism

Experimental studies have shown that the anti-cancer activity of Se is dependent on the chemical form affecting its intracellular distribution in the body [21,22]. Se in the diet may be delivered in organic or inorganic forms and the most common dietary Se compounds are selenomethionine (SeMet), selenocysteine (SeCys), Se-methylselenocysteine (MeSeCys), and selenite [23,24] (Figure 2). SeMet is the dominant form in food products [25] with vegetables, grains, legumes, nuts, and yeast rich in the organic forms of Se. Inorganic Se is also observed in some of these foods and water [23,26,27], with SeMet found in many food products, including grains, meat, eggs, dairy products, nuts (especially Brazil nuts), and yeast [28]. Dietary sources of SeCys are animal products [29], whereas *Brassica* and *Allium* species, including garlic and onions, are sources of MeSeCys. Selenite occurs naturally in foods in small amounts, and the main source of the inorganic Se compounds in the human diet is supplementation [30].

The anti-cancer activity also depends on the metabolism of the Se compound [31], with diverse metabolic pathways producing various Se metabolites [32,33] (Figure 3). This is particularly relevant in cancer prevention as the biological activities of the Se compounds are mainly exerted via their metabolites [34,35].

Selenate is the inorganic form of Se which is present in small amounts in both plant and animal products [36]. Selenate and selenite are reduced to hydrogen selenide (H_2_Se) through an intermediate form selenogultatione (GSSeSG) using NADPH (nicotinamide adenine dinucleotide phosphate) and glutathione reductase [21,37]. The H_2_Se is phosphorylated to selenophosphate (H_3_SePO_3_) from which SeCys is formed for selenoprotein biosynthesis [38].

Twenty-five selenoproteins have been identified in humans so far, most of which have an as-yet-unknown function [39,40,41], but some may have anti-cancer effects [42]. Dietary SeCys is transformed into hydrogen selenide through the β-lyase reaction [43]. SeMet is converted to SeCys by trans-selenation [44] or to methylselenol (MeSeH) by cystathionine γ-lyases [45,46,47] and is mainly involved in the production of selenoproteins, while the alternative γ-lyase pathway plays only a minor role [32]. Se-methylselenocysteine is transformed to MeSeH via β-lyase, and then MeSeH is further methylated to dimethylselenide (DMeSe) and trimethylselenide (TMeSe). Methylated selenides are the end products of Se metabolism that are excreted from the body by urine or breath [48]. Since MeSeCys can be converted directly to methylselenol, this explains why they may be more efficacious than other Se forms in cancer prevention [49,50]. MeSeH is also used to produce selenosugars which are excreted in the urine [51]. Moreover, MeSeH can be demethylated to selenide for further conversion to selenoproteins [43] and has been shown to have anti-tumour effects [52,53,54,55].

The Se anti-cancer properties are not fully understood but may be influenced by antioxidant protection, enhanced immune surveillance, modulation of cell proliferation, inhibition of angiogenesis, and tumour cell invasion [56,57]. Additionally, Se may reduce the incidence of cancer through its effect on apoptosis [49,58,59]. Se-induced apoptosis is not fully understood but the complex mechanisms may include ROS generation, protein kinases signalling, activation of caspases, and p53 phosphorylation [60].

### Selenium Bioavailability

Se bioavailability depends on the source and chemical forms of the element [61]. Se content in foods is determined by combination of geologic and environmental factors and Se supplementation of fertilisers and animal feedstuffs [33]. Chemical forms of Se differ in absorption and conversion into a biochemically active form. Organic sources are assimilated more efficiently and are considered to be less toxic than inorganic compounds [27].

The bioavailability of selenium may also be affected by some other components of the food matrix [62]. Some studies have shown that vitamin E can increase the bioavailability of selenium, while heavy metals and fibre decrease it [63,64,65]. Dietary sulphur (especially from methionine) may compete with Se for absorption. Additionally, bioavailability may influence parameters related to the human body, i.e., age, sex, Se status, or lifestyle [66].

Current research is concerned with improving bioavailability and identification of biomarkers of exposure and anti-cancer activity. Interestingly, in recent years, studies have shown that some Se enrichment of yeasts and lactic acid bacteria represent sources of more bioavailable organic and less toxic forms of Se [67].

## 3. Functional Foods Rich in Selenium

According to the European Food Safety Authority (EFSA), “a functional food is defined as a food, which beneficially affects one or more target functions in the body, beyond adequate nutritional effects, in a way that is relevant to either an improved state of health and well-being and/or reduction of risk of disease. A functional food can be a natural food or a food to which a component has been added or removed by technological or biotechnological means, and it must demonstrate its effects in amounts that can normally be expected to be consumed in the diet” [68].

The Functional Food Center (FFC) defines a functional food as a natural or processed food that contains known or unknown biologically active compounds, which in defined, effective, non-toxic amounts provide clinically proven and documented health benefits in the prevention and treatment of chronic diseases [69,70].

In the United States, functional foods are regulated in the same way as conventional foods and dietary supplements. The primary distinction between functional food and food in general is in the claims made for benefits, other than nutritional benefits, attributed to the functional food [71].

Unfortunately, most countries do not have a formal definition of functional food, and therefore terms such as dietary supplements, nutraceuticals, or medical foods are often used. Consequently, functional food may be defined as natural food enriched with a health-promoting ingredient, food where some component has been added for special health reasons, or food where the ingredient has been technologically or chemically modified [72,73]. Accordingly, functional food rich in Se for prevention can be classified into natural Se-rich food products, foods fortified with Se, and supplements.

### 3.1. Food Products Naturally Rich in Selenium Used in Cancer Prevention

The amount of Se in foods varies and depends upon geological and geographical factors [74] (Table 1). For example, the Se content in plants depends upon the type of soil and its natural Se content, the use of Se-enriched fertilisers, and Se bioavailability [75]. Depending on the ability to accumulate Se, plants may be classified as non-Se-accumulators, secondary Se-accumulators, and Se-accumulators. Se-accumulators can contain up to 40,000 µg Se/g when grown in Se-rich environments, while non-accumulators rarely collect more than 100 µg Se/g dry weight. The only Se-accumulator plant used as a food source is *Bertholletia excelsa*, the origin of Brazil nuts, with some species of *Brassica* and *Allium* being secondary Se-accumulators.

The Se content in animal products is much lower and depends primarily on the Se content of the feed. Most Se is found in fish (0.4−4.3 µg/g), organs such as the liver and kidney (0.2−2.0 µg/g), and muscles (around 0.3 µg/g) [27]. Moreover, the food preparation process affects the Se, with cooking in water reducing Se by 5–50%, depending on the type of product [80,81].

### 3.2. Brazil Nuts

Brazil nuts obtained from the *Bertholletia excelsa* tree belong to the *Lecythidaceae* family and are native to South America [82,83]. They are good sources of nutrients, including protein, fibre, vitamins, minerals, and other bioactive compounds, and thus have various potential health benefits [84,85]. Brazil nuts are the richest known food source of Se, with an average concentration of 2 to 20 µg/g [76]. However, concentrations in individual nuts vary considerably (0.03−512 µg/g) [86] depending on the Se content and bioavailability in the soil [87,88]. Se bioavailability in Brazil nuts is the same as in selenite, which is used to restore Se activity in tissues and selenoprotein; therefore, it is believed to be helpful in cancer prevention [89] (Table 2). Ip et al. (1994) [90] showed that Se compounds in Brazil nuts maintain the activity of selenoenzymes (glutathione peroxidase and type I 5′-deiodinase), suggesting that the Se bioavailability in Brazil nuts is similar to that of selenite. In rats, Brazil nuts as a source of Se (16 and 30 µg/g) were as effective as a similar amount of selenite in maintaining selenoenzyme activity and preventing mammary cancer.

A beneficial anti-cancer prevention effect has also been observed in humans [92]. Thomson et al. (2008) [89] showed that 12 weeks of consuming 2 Brazil nuts (average 53 µg Se) per day increased the plasma Se concentration and enhanced glutathione peroxidase (GPx) activity. In an epidemiological observational study, the effect of 6 weeks of supplemented Brazil nuts (6 nuts per day, 48 µg Se) and green tea extract (800 mg of epigallocatechin-3-gallate) reduced the risk of colorectal cancer by regulating genes associated with selenoproteins, WNT signalling (β-catenin), inflammation (NF-κB), and methylation (DNMT1). However, the interventions did not significantly affect the plasma CRP (C-reactive protein) level and rectal acetylated histone H3 or Ki-67 expression. Furthermore, the combination did not provide additional effects compared to each agent separately [91].

The beneficial anti-cancer effect of Brazil nuts may be related not only to their high Se content but also to other nutrients, e.g., polyphenols. Studies by Yang et al. evidenced the proliferation of HepG2 and Caco-2 cell lines, which were significantly inhibited in a dose-dependent manner after exposure to nut extracts [93]. Even though Brazil nuts are a convenient source of Se in the human diet, care should be taken to avoid the possible toxic effects associated with a chronically high Se, Ba, and Ra intake. Therefore, the intake should be limited to 30 g of nuts per day (around 5–6 nuts) [76,82,90,94].

It is worth noting the toxic values of barium and radium taken with food is not clearly established; however, Ba toxicity has been reported with ingestions as small as 200 mg Ba/kg/day [95,96].

### 3.3. Brassica Species

The *Brassicaceae* family is a large group with around 3000 species and occurs worldwide [97]. The most commonly consumed species are broccoli, brussels sprouts, cauliflower, and cabbage [98]. *Brassica* species have a high capacity to accumulate Se as it can replace Se in the proteins, although this accumulation is limited by soil Se concentrations [78,99] and is around 0.029−0.247 µg/g^−1^ in dry mass (turnip 0.029 μg/g, kale 0.046 μg/g, cauliflower 0.102 μg/g, broccoli 0.129 μg/g, and brussels sprouts 0.247 μg/g) [78]. Several members of the *Brassicaceae* family accumulate up to 10 µg/g dry mass in their tissues when grown in Se-rich soil, including broccoli, kale, and cabbage [100]. Se compounds such as MeSeCys and selenoglucosinolates in *Brassica* tissues may have a potential anti-cancer activity [101].

In vivo and in vitro studies have confirmed the anti-cancer activity of Se-enriched broccoli and broccoli sprouts in colon, mammary, intestinal, and prostate cancers. Finley et al. reported that 3 weeks of a diet of salinised broccoli in rats significantly decreased the incidence of aberrant preneoplastic lesions (indicative of colon cancer), aberrant crypts, and aberrant crypt foci. Additionally, the high-Se broccoli was more effective than selenate or selenite for cancer prevention, probably related to the unique chemical form of Se in broccoli (MeSeCys) [102]. Similar results were observed for Se-fortified brussels sprouts on mammary cancer cells [103]. Davis et al. [104] fed mice with Se-enriched broccoli (2.1 mg Se/kg in diet) and noted significantly fewer small intestinal tumours and a smaller total tumour burden than in mice fed the control diet (0.11 mg Se/kg in diet).

Moreover, Zeng et al. showed that the Se-enriched broccoli activates specific pro-apoptotic genes linked to tumour protein p53, activator protein 1, nuclear factor kappa-light-chain-enhancer of activated B cells, and stress signal pathways in response to tumour formation [105]. Tsai et al. [106] observed that Se-enriched broccoli extract inhibits the growth of HCT116 and HCT116+Chr.3 human colon cancer cells. Se-enriched Japanese radish sprouts, kale, and kohlrabi sprouts also showed cancer prevention activity; however, there is less research available on these vegetables. A diet of Se-enriched Japanese radish sprouts in rats caused a significantly lower incidence of mammary tumours compared to the control group [107]. The Se-enriched kale and kohlrabi sprout extracts (≥1 mg/mL) showed cytotoxic potency to all the studied SW480, SW620, HepG2, and SiHa human metastatic cancer cell lines [108]. Detailed information on studies on *Brassica* species in cancer prevention is presented in Table 3.

Vegetables from the *Brassica* family are characterised by potential anti-cancer activity; however, their beneficial effect is dependent on their Se content. In the review by Ramires et al. (2020) [110], it was noted that functional food, such as broccoli powder or kale powder, was available on the market, but was relatively expensive. Powdered vegetables can potentially contain higher amounts of Se but labels lack information about the Se content.

### 3.4. Allium Species

*Allium* species belongs to the *Alliaceae* family and are one of the oldest cultivated vegetables used as food [111]. The most popular *Alliums* widely used all over the world are garlic (*Allium sativum*) and onion (*Allium cepa*) [112]. Some in vitro and in vivo studies showed *Allium* species, especially garlic, have an anti-cancerogenesis effect [113,114,115]. It has been suggested that the chemopreventive effect of vegetables is due to the various active compounds, especially organosulphur, organoselenium, and polyphenols [116,117,118,119,120,121] (Table 4). The differences in the anti-cancer effect may be related to the dose of active substances and processing losses such as thermal treatment [113].

The Se concentration in regular *Allium species* is <0.5 µg/g, while in Se-enriched garlic and onion, Se concentrations in dry mass vary from 68−1355 µg/g and 96−601 µg/g, respectively [26,33,126,127,128]. Obviously, the concentration of Se in Se-enriched vegetables is dependent on the intensity of Se fertilisation [50].

There are two dominant forms of Se in selenised garlic, MeSeCys and γ-glutamyl-MeSeCys [129]. In humans, MeSeCys and γ-glut-MeSeCys enter the methylated pool of Se and are transformed into MeSeH by β-lyase. It has been shown that for MeSeCys, the production of a monomethylated Se metabolite from MeSeCys via β-lyase is a key step in its cancer chemoprevention [130]. As methylselenol has been recognised as an anti-cancer compound, the consumption of foodstuff containing this precursor of MeSeH is generally recommended [62].

Natural allium and that cultivated with Se fertilisation have a protective role in cancer prevention [113,129]. However, Se-enriched garlic is more anti-carcinogenic than regular garlic or other chemically defined sources of Se such as selenite or SeMet [131,132]. Furthermore, the anti-cancer activity of Se-enriched garlic with a moderate Se content (100–300 µg/g Se dry weight) was very similar to that of garlic with a high Se content (>1000 µg/g Se) [50].

The chemopreventive effect of Se-enriched allium vegetables has been observed in mammary cancer. Ip et al. showed that garlic cultivated with selenite fertilisation has a chemopreventive activity in the rat 7,12-dimethylbenz[a]anthracene (DMBA)-induced mammary tumour model [122]. Furthermore, consumption of Se-enriched garlic and Se-enriched onion does not cause excessive Se accumulation in tissues with no observed perturbation in the maintenance of functional selenoenzymes even at a high dose of supplementation [123]. Moreover, Se-enriched garlic is more potent than Se-enriched yeast in suppressing the growth of premalignant lesions and the formation of adenocarcinomas in the mammary gland of carcinogen-treated rats [125].

However, Allium vegetable consumption may be limited by personal preferences as well as their characteristic taste and smell. In addition, to meet the Se demand, a much larger amount of natural garlic than that enriched with Se must be consumed [133].

### 3.5. Se-Enriched Yeast

Se-enriched yeast is a common form of Se used for dietary supplements [134]. Production of Se-enriched yeast is more manageable than the production of other Se-enriched foods (e.g., Se-enriched broccoli or Se-enriched onion) and less costly [135,136]. It is produced by the anaerobic fermentation of *Saccharomyces cerevisiae* in a Se-enriched medium [137]. The main seleno compound in Se-enriched yeast is selenomethionine, representing approximately 60–85% of the total Se [134,138,139], followed by selenocysteine (approximately 2–4% of all types of Se). The Se content in the commercial dried product does not exceed 2.5 mg Se/g (range 1−2.4 mg Se/g) [139].

It has been suggested that the use of Se-enriched yeast, in the case of deficiency, has a multidirectional beneficial effect on human health [140]. However, the anti-cancer effect of Se-enriched yeast is controversial. Some studies confirmed the beneficial effects of Se-enriched in cancer [134,141], but the use of Se-enriched yeast in the prevention of prostate cancer development did not prove beneficial and in some studies increased the risk of developing cancer [142,143,144].

The effect of Se-enriched yeast on chemoprevention in humans is shown in Table 5. Clark et al. (1996), in a double-blind, randomised, placebo-controlled study, observed the effect of supplementation with Se-enriched yeast on skin cancer. Patients treated for 4.5 (mean 2.8) years with 50 µg of Se-enriched yeast (200 µg of Se) were not protected against the development of basal or squamous cell carcinoma of the skin. However, the analysis of secondary endpoints supports the hypothesis that Se supplementation may reduce the incidence of lung, colorectal, and prostate cancer [7]. In a re-analysis of the data to include a further 25 months of blinded intervention, Se-enriched yeast supplementation significantly reduced total and prostate cancer incidence [134,145]. Yu et al. [141] observed no primary liver cancer after 4 years of supplementation with selenised yeast (200 µg Se/day) compared to the placebo group.

Current human clinical trial results have not demonstrated a chemopreventive effect of Se-enriched yeast on prostate cancer cells. El-Bayoumy et al. reported that 9 months of Se-enriched yeast supplementation increased blood glutathione levels and significantly decreased prostate-specific antigen levels in the Se-enriched yeast group [144]. Algotar et al. found no significant differences in the time to prostate cancer diagnosis after several years of supplemented high-Se yeast (200 µg and 400 µg Se/day).

### 3.6. Other Se-Enriched Food Products

Other functional Se-enriched products include mushrooms, tea, and milk. Their anti-tumour activity has been demonstrated in lung, breast, colorectal, and murine sarcoma cancer cells (Table 6).

Hu et al. showed that Se-enriched milk (1 μm/g) may protect against colorectal cancer in a mouse model, with the upregulation of GPx-2 and selenoproteins in the colon crucial for chemoprevention [147]. A subsequent study of 26 healthy human participants showed rectal selenoprotein gene expression (i.e., SeP, GPx-1 and GPx-2) is significantly regulated by dietary Se supplementation independent of Se plasma levels and GPx activity. This regulation depends on the form of dietary Se supplementation, with Se-enriched milk having a more sustained effect than Se-enriched yeast [148].

In an in vitro study, certain species of mushrooms such as *Ganoderma lucidum* used as traditional medicine in Asia have potential anticancer effects in non-small-cell breast cancer [149]. The potential anticancer activity was also observed in non-small-cell lung cancer for other mushroom species such as *Cordyceps militaris* [152]. The anticancer activity against lung cancer cells was observed with *Ulva fasciata* also known as sea lettuce, a common green alga consumed in many parts of the world [150].

Cheng et al. showed that Se-containing tea polysaccharides (Se-TPS) from Se-enriched tea significantly inhibited the proliferation of S-180 cells in a dose-dependent manner. In an animal model, Se-TPS oral administration effectively inhibited tumour growth in a dose-dependent manner. The anti-tumour effect of Se-TPS was significantly higher than that of tea polysaccharides and Se-yeast due to the synergistic effect of Se and tea polysaccharides [151].

## 4. Concluding Remarks—The Chemoprevention Effect of Se-Enriched Products

In vitro and animal studies confirmed the beneficial chemoprevention effect of functional products rich in Se; however, this was not confirmed in clinical trials. Consequently, future clinical research should determine the appropriate dosage and form of Se in food products as well as use an appropriate study sample (determine their dietary Se intake, effect of processing, and solubility of Se and exclude influencing factors, e.g., gender). It is of note that the chemopreventive dose of Se (200 µg/day) is much higher than the adequate intake (70 µg/day for adults) and is near the upper limit for this element (255 µg/day), and therefore control of human intake is required. A summary of the chemoprevention effects of Se-enriched products is shown in Table 7.

## 5. Conclusions

Se may have the potential to prevent cancer but most evidence to date is from animal studies. The basis for future research on the validity of the introduction of Se as a preventive factor in humans may be randomised control and epidemiological studies conducted in humans. The beneficial anti-cancer properties of Se and the difficulty in meeting the daily requirements of this micronutrient in some populations make it crucial to continue the research. However, the question remains: what is the optimal form of Se for chemoprevention and what biomarkers should be selected to predict those who will derive the most benefit? Additionally, further research should focus on finding products characterised by more bioavailable and less toxic forms of Se within the context of the anti-cancer activity.

## Figures and Tables

**Figure 1 foods-12-01247-f001:**
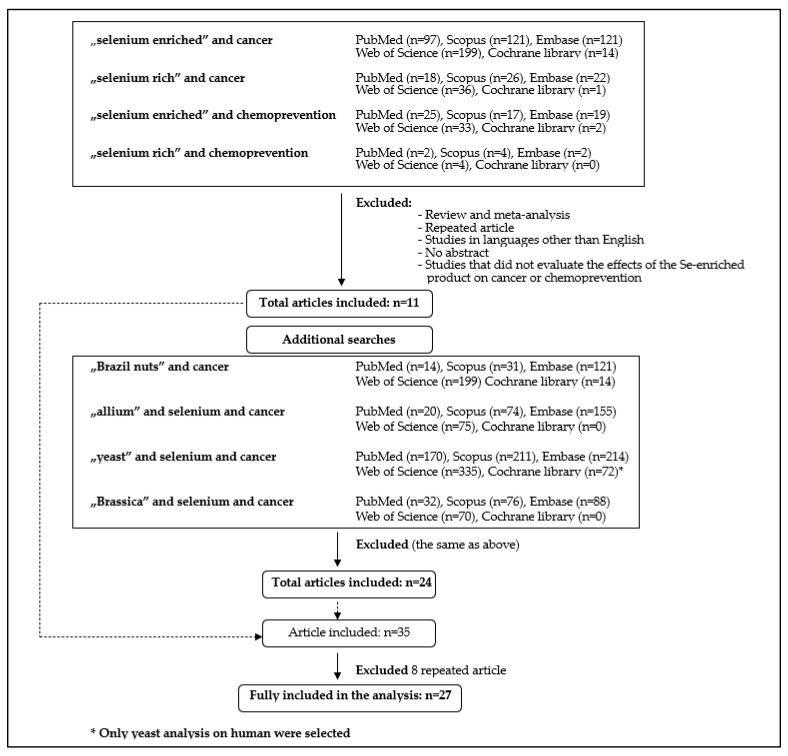
Search and selection methodology of articles used in this review.

**Figure 2 foods-12-01247-f002:**
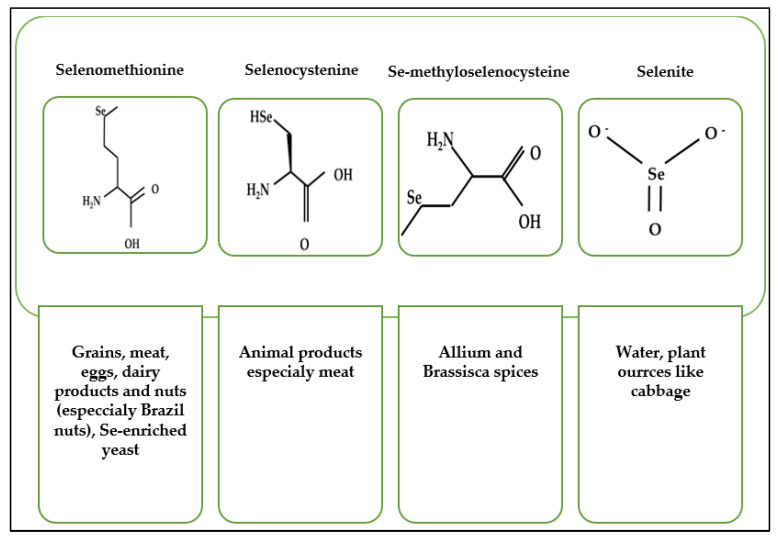
Dietary sources of selenium compounds.

**Figure 3 foods-12-01247-f003:**
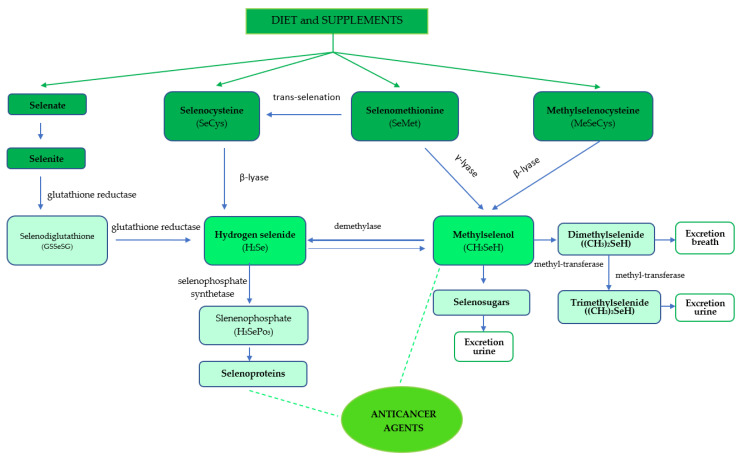
The metabolic pathways of selenium compounds in the human body.

**Table 1 foods-12-01247-t001:** Main food sources of selenium.

Food Source	Average Content µg/g	Reference
Brazil nuts	2−20	Parekh P. P. et al., 2008 [76]
Garlic	0.15	Larsen E. H. et al., 2006 [77]
Broccoli	0.13 *	De Temmerman et al., 2014 [78]
Brussels sprouts	0.25 *	De Temmerman et al., 2014 [78]
Fish	0.4−4.3	Rayman M. P.et al., 2008 [27]
Meats (mussels)	0.3	Rayman M. P. et al., 2008 [27]
Yolk from egg	0.12−0.42	Pilarczyk B. et al., 2019 [79]

* Se content in dry mass.

**Table 2 foods-12-01247-t002:** Effect of Brazil nuts on cancer prevention.

Form, Sources and Dose of Se	Period	Effect	Experimental Model	Reference
Two preparations of processed Brazil nuts Experiment 1: Final dietary Se levels 2 and 3 µg/g (from Brazil nuts content 16 µg Se/g. Experiment 2: Final dietary Se levels 1 and 2 µg/g (from Brazil nuts 30 µg Se/g) (dominant form of Se: MeSeCys) vs. Selenite—dietary Se content 1 and 2 µg/g	2 weeks and 6 months (2 weeks before administration of 7,12-dimethylbenz(a)anthracene and 6 months after administration)	Mammary cancer: protection increased Se retention in the mammary gland, plasma, liver, and kidney Se in Brazil nuts and selenite are similarly bioactive	Pathogen-free female Sprague-Dawley rats	Ip C. et al., 1994 [90]
Two Brazil nuts Average 53 µg/g Se per day (possible range: 20–84 µg Se) Dominant form of Se: SeMet	14 weeks	Increased plasma Se concentration and enhanced GPx activity	59 New Zealand adults	Thomson C. D. et al., 2008 [89]
Brazil nuts and green extract, alone and in combination Six Brazil nuts Average 48 µg/g Se per day	6 weeks	Colorectal cancer: regulated genes associated with selenoproteins, WNT signalling (β-catenin), inflammation (NF-κB), and methylation The combination of Brazil nuts and green extract does not provide additional effects compared with either agent alone	61 adults aged 52–75 years	Hu Y. et al., 2016 [91]

MeSeCys, Se-methylselenocysteine; SeMet, selenomethionine.

**Table 3 foods-12-01247-t003:** Effect of *Brassica* vegetables on cancer prevention.

Form, Sources and Dose of Se	Period	Effect	Experimental Model	Reference
Se-enriched broccoli 2 μg Se/g diet as high Se broccoli (selenised broccoli—Se concentration 500 µg/g)	3 weeks	Colon cancer: decreases aberrant crypts and aberrant crypt foci	Fisher F-344 inbred rats, weanling males	Finley J. W. et al., 2000 [102]
Se-fertilised broccoli or broccoli florets First experiment: 3 μg of Se/g of diet Second experiment: 2 µg Se/g of diet	22 weeks	Mammary cancer: significantly reduces the number of tumours Colon cancer: significantly fewer aberrant colon crypts	Pathogen-free female Sprague–Dawley rats	Finley J. W. et al., 2001 [103]
Se-enriched broccoli sprout (dominant form of Se: MeSeCys) 2.1 mg Se/kg diet or 0.11 mg Se/kg (control group)	10 weeks	Intestinal cancer: small tumours and a smaller total tumour burden	Heterozygotic male Min mice	Davis C. D. et al., 2002 [104]
Se-enriched broccoli (dominant form of Se: MeSeCys) 2.1 mg Se/kg diet or 0.11 mg Se/kg (control diet)	10 weeks	Intestinal cancer: activates specific pro-apoptotic genes linked to tumour	Heterozygotic male Min mice	Zeng H. et al., 2003 [105]
Se-enriched Japanese radish sprouts (dominant form of Se: MeSeCys) 8.8 µg of Se/in diet or under 1 µg of Se/g in a control diet	13 weeks	Mammary cancer: significantly lowers the incidence of tumours in the Se-enriched sprout-added test diet group than in the basal diet group	Virgin female Sprague–Dawley strain rats	Yamanoshita O. et al., 2007 [107]
Se-enriched broccoli sprouts (dominant form of Se: MeSeCys) 24.2 μM	72 h	Prostate cancer: inhibits cell proliferation, decreases prostate-specific antigen secretion, and induces apoptosis	In vitro model	Abdulah et al., 2009 [109]
Se-enriched broccoli extract 1.08 × 10^−4^ M Se and 2.50 × 10^−7^ M Se	2 × 48 h	Colon cancer: inhibits the growth of HCT116 and HCT116+Chr.3 human colon cancer cells	In vitro model	Tsai C. F. et al., 2013 [106]
Se-fortified kale and kohlrabi sprouts (dominant form of Se: SeMet) 0.07−0.17 mg of Se/g dried weight	24 h	Human metastatic cancer: cytotoxic effect on SW480, SW620, HepG2, SiHa cell lines (at ≥1 mg/mL sprouts extract)	In vitro model	Zagrodzki P. et al., 2020 [108]

MeSeCys, Se-methylselenocysteine; SeMet, selenomethionine.

**Table 4 foods-12-01247-t004:** Effect of *allium* vegetables on cancer prevention.

Form, Sources and Dose of Se	Period	Effect	Experimental Model	Reference
Se-enriched garlic 150 µg/g Se (concentration in diet 3 µg/g Se) vs. Regular garlic 0.06 µg/g Se (concentration in diet 0.0012 µg/g Se) vs. Selenite 3 µg/g Se (in diet 3 µg/g Se) vs. Control group (dominant form of Se in Se-enriched garlic: MeSeCys)	26 weeks	Mammary cancer: inhibited total tumour yield and tissue Se levels were lower in animals ingesting the Se-enriched garlic than selenite	Pathogen-free female Sprague–Dawley rats	Ip C. et al., 1992 [122]
Experiment I - Control (0.1 µg/g Se) - 0.85% regular garlic (0.1 µg/g Se) - 1.7% regular garlic (0.1 µg/g Se) 0.85% high Se garlic (1 µg/g Se) - 1.7% high Se garlic (2 µg/g Se) Experiment II - Control (0.1 µg/g Se) - 3.5% regular onion (0.1 µg/g Se) - 7% regular onion (0.1 µg/g Se) 3.5% high Se onion (1 µg/g Se) - 7% high Se onion (2 µg/g Se) (dominant form of Se in Se-enriched garlic: MeSeCys)	8 months and 2 weeks	Mammary cancer: consumption does not cause excessive Se accumulation in tissues	Female Sprague–Dawley rats	Ip C. et al., 1994 [123]
Se-enriched garlic 112 µg/g and 1355 µg/g in dry weight (final concentration in diet 2 µg/g Se) vs. Control group 0.1 µg/g Se in diet (dominant form of Se in Se-enriched garlic: MeSeCys)	3 weeks and 22 weeks	Mammary cancer: tumour reduction was due to the effect of Se not the effect of garlic	Pathogen-free female Sprague–Dawley rats	Ip C. et al., 1995 [124]
Se-enriched garlic diet concentration 3 µg/g Se vs. Sodium selenite diet concentration 3 µg/g Se vs. Control group 0.01 µg/g Se in diet (dominant form of Se in Se-enriched garlic: MeSeCys)	7 weeks	Mammary cancer: inhibited total tumour yield, as well as the proliferation, survival and matrix degradation of endothelial cells critical for angiogenesis	Female Sprague–Dawley rats	Jiang C. et al., 1999 [121]
Se-enriched garlic (296 µg/g Se) and Se-enriched yeast (1922 µg/g Se) (final concentration in diet 1, 2 or 3 µg/g Se) vs. Control group 0.1 µg/g Se in diet (dominant form of Se in Se-enriched garlic: γ-glutamyl-MeSeCys)	6 weeks	Mammary cancer: decreased morbidity and mortality Se-garlic was significantly more effective in suppressing the development of premalignant lesions and adenocarcinomas than Se-yeast despite Se-enriched yeast having a higher total tissue Se content	Pathogen-free female Sprague–Dawley rats	Ip C. et al., 2000 [125]

MeSeCys, Se-methylselenocysteine; γ-glutamyl-MeSeCys, γ-glutamyl-Se-methylselenocysteine.

**Table 5 foods-12-01247-t005:** Effect of Se-enriched yeast on cancer prevention in human studies.

Form, Sources and Dose of Se	Period	Effect	Experimental Model	Reference
High-Se brewer’s yeast tablet (200 µg Se/day) vs. Placebo	4.5 ± 2.8 years	Skin cancer: Se supplementation does not protect against the development of basal or squamous cell carcinomas but may reduce total cancer, lung, colorectal, and prostate cancer incidence, as well as lung cancer mortality	1312 patients with a history of basal cell or symptoms of cell carcinomas, randomly assigned to the Se-treatment group (n = 653) and placebo group (n = 659); Mean age 63 years	Clark L. C. et al., 1996 [7] and Combs G. F. et al., 1997 [146]
Selenised yeast tablets 200 µg Se/day vs. Placebo	4 years	Primary liver cancer: no primary liver cancer was observed in 113 people supplemented with Se during the 4-year study, while 7 of the placebo group were diagnosed with primary liver cancer	226 participants randomly assigned to the study group (n = 113) and placebo group (n = 113); aged 21−63 years	Yu S. Y. et al., 1997 [141]
Se-enriched yeast 247 µg Se/day vs. Placebo	9 months + 3 months placebo after supplementation	Prostate cancer: increased blood glutathione levels and significantly decreased prostate-specific antigen levels	36 healthy adults randomly assigned to the study group (n = 17) and placebo group (n = 19); aged 19−43 years	El-Bayoumy K. et al., 2002 [144]
High-Se yeast (200 or 400 µg Se/day) vs. Placebo	patients in the United States—5 years, patients in New Zealand no more than 3 years	Prostate cancer: no significant differences in the time to prostate cancer diagnosis between placebo and study groups	699 men at high risk for prostate cancer randomly assigned to 200 µg Se/day (n = 234), 400 µg Se/day (n = 233) or placebo group (n = 232); aged < 80 years	Algotar A. M. et al., 2013 [143]

**Table 6 foods-12-01247-t006:** Effect of Se-enriched food products on cancer prevention.

Form, Sources and Dose of Se	Period	Effect	Experimental Model	Reference
Se-enriched milk proteins vs. Se-enriched yeast (dominant form of Se: SeMet) Four groups: - milk protein control diet (0.068 µg/g Se) - dairy-Se diet (0.5 µg/g Se) - dairy-Se diet (1 µg/g Se) - milk protein control + yeast-Se diet (1 µg/g Se)	4 weeks	Colorectal cancer: Se-enriched milk regulates colonic GPx-2 and selenoprotein P mRNA expression	Male C57BL/6J mice	Hu Y. et al., 2010 [147]
Se-enriched milk protein (150 μg/d) vs. Se-enriched yeast (150 μg/d) (dominant form of Se: SeMet)	6 weeks	Colorectal cancer: selenoprotein gene expression (selenoproteins P, GPx-1, GPx-2) was regulated by dietary Se independent of plasma Se levels, and GPx activity Se-enriched milk had a more sustained effect than Se-enriched yeast	23 healthy volunteers randomly assigned to consume Se-enriched milk (n = 12) or Se-enriched yeast (n = 11); aged 52−79 years	Hu Y. et al., 2011 [148]
Se-enriched *Ganoderma lucidum* 0.045 to 0.36 μM SeGLP-2B-1	24, 48 or 72 h	Breast cancer: inhibited the growth of breast cancer cells in a time- and dose-dependent manner and increased caspase-9 and caspase-3 activity	MCF-7 human breast cancer cells	Shang D. et al., 2011 [149]
Se-enriched *Ulva fasciata* A549 cells were treated with 3, 4, 5 and 6 µg/mL	72 h	Lung cancer: induced apoptosis (sub-G1 phase cells, upregulation of p53, and activation of caspase-3 in lung cancer cells)	A549 human lung cancer cells	Sun X et al., 2017 [150]
Se-containing tea polysaccharides (Se-TPS) from Se-enriched tea vs. Se-enriched yeast (Se 89 µg/g) Six groups: - control - Se-yeast 100 mg/kg - TPS 100 mg/kg - Se-TPS 50 mg/kg - Se-TPS 100 mg/kg - Se-TPS 200 mg/kg	13 days	Murine sarcoma (S-180): Se-TPS significantly inhibited the proliferation of S-180 cells in a dose-dependent manner; in vivo, Se-TPS inhibited tumour growth in a dose-dependent manner	In vitro model and Kunming mice	Cheng L. et al., 2018 [151]
Se-enriched *Cordyceps militaris* NCI-H292 cells were treated with 0, 4, 8 and 12 mg/mL and A549 cells were treated with 0, 12.5, 25 and 50 mg/mL	24 h	Lung cancer: cell poliferation, apoptosis in non-small cell lung cancer	Human lung cancer cell lines NCI-H292 and A549	Luo L. et al., 2019 [152]

SeMet, selenomethionine.

**Table 7 foods-12-01247-t007:** Summary of the chemoprevention effects of Se-enriched products.

Products	Potential Chemopreventive Effect	Availability on the Market	Recommended Daily Dose *	Comments/References
Brazil nuts	Mammary and colorectal cancer	Easily accessible	4 nuts ^1^	- The Se content depends on the origin of the nuts - Excess may lead to toxic effects [89,94,153]
Se-enriched broccoli	Mammary, colon, intestinal, prostate, human metastatic cancer	Not yet available for sale Only dried broccoli powder with unspecified Se content is available	Probably Se-enriched broccoli powder 0.15−1 g (depending on the enrichment process)	- The Se content depends on the enrichment process - No human clinical trials with Se-enriched broccoli
Se-enriched garlic	Mammary cancer	Not yet available for sale Only powdered garlic with unspecified Se content is available	Probably Se-enriched broccoli powder 0.15−1 g (depending on the enrichment process)	- The Se content depends on the enrichment process - No human clinical trials with Se-enrich garlic
Se-enriched yeast	Skin cancer, primary liver cancer	Easily accessible	Around 0.5 g (depending on the enrichment process)	- No significant effect on prostate cancer in a clinical human study

* 200 µg/day was adopted as the chemopreventive dose [7,19]. ^1^ The average Se content in Brazil nuts was assumed to be 10 µg/g, and one nut has an average of 5 g.

## Data Availability

No new data were created or analyzed in this study. Data sharing is not applicable to this article.

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
