# Peer review of "Natural Sources of Selenium as Functional Food Products for Chemoprevention"

_foods, 2023, doi:10.3390/foods12061247_

Round 1

Reviewer 1 Report

Line 117: selenenoproteins should be changed to selenoproteins

Line 129: Section 3 should be improved with an overview of definitions/regulations from other countries/areas. 

Line 190: This sentence should be improved. I suggest: Even though Barzil nuts are convenient source of Se in the human diet, care should be taken to avoid the possible toxic effects associated with a chronically high Se, Ba and Ra intake. Therefore, the intake should be limited to 30 g of nuts per day (around 5-6 nuts).

Line 191: What are the toxic values of Ba and Ra for humans? These data should be added.

Line 223: In this line you are mentioning HCT116 two times. Please check if this is or not a mistake.

-Please uniform trough the text the symbol for micro. Sometimes you use u, and sometimes μ.

Author Response

Response to the comments made by the reviewers

Manuscript ID: foods-2267164

Title: Natural sources of selenium as functional food products for chemoprevention

We would like to thank the Reviewers for their careful review of our manuscript and for providing us with some suggestions to improve its quality. We have carried out a major revision of the manuscript, and we believe the paper has improved significantly.

According to the Reviewers' suggestion, the manuscript has been carefully checked and corrected. The changes in the manuscript have been highlighted in red.

Below we sequentially address all of the points raised by the Reviewers.

Reviewer 1:

Firstly, we would like to express our profound thanks to the Reviewer for devoting time to reviewing our manuscript, the corrections and suggestions. We have carried out a major revision of the manuscript, and we believe the paper has improved significantly.

The Reviewer's comment: Line 117: selenenoproteins should be changed to selenoproteins

The authors' answer: According to the Reviewer's suggestion, we have been corrected it.

The Reviewer's comment: Line 129: Section 3 should be improved with an overview of definitions/regulations from other countries/areas.

The authors' answer: According to the Reviewer's suggestion, we have filled in the missing information into the manuscript and highlighted in red.

According to the European Food Safety Authority (EFSA), “a functional food is defined as a food, which beneficially affects one or more target functions in the body, beyond adequate nutritional effects, in a way that is relevant to either an improved state of health and well-being and/or reduction of risk of disease. A functional food can be a natural food or a food to which a component has been added or removed by technological or biotechnological means, and it must demonstrate their effects in amounts that can normally be expected to be consumed in the diet” [68].

The Functional Food Center (FFC) defines functional food as a natural or processed food that contains known or unknown biologically active compounds, which in defined, effective, non-toxic amounts provide clinically proven and documented health benefits in the prevention and treatment of chronic diseases [69,70].

In United States functional foods are regulated in the same way as conventional foods and dietary supplements. The primary distinction between functional food and food in general is in the claims made for benefits, other than nutritional, attributed to the functional food [71].

The Reviewer's comment: Line 190: This sentence should be improved. I suggest: Even though Brazil nuts are convenient source of Se in the human diet, care should be taken to avoid the possible toxic effects associated with a chronically high Se, Ba and Ra intake. Therefore, the intake should be limited to 30 g of nuts per day (around 5-6 nuts).

The authors' answer: All changes have been made in the manuscript and highlighted in red.

Even though Brazil nuts are convenient source of Se in the human diet, care should be taken to avoid the possible toxic effects associated with a chronically high Se, Ba and Ra intake. Therefore, the intake should be limited to 30 g of nuts per day (around 5-6 nuts) [78,82,90,94].

The Reviewer's comment: Line 191: What are the toxic values of Ba and Ra for humans? These data should be added.

The authors' answer: According to the Reviewer's suggestion, we have filled in the missing information into the manuscript and highlighted in red.

It is worth noting the toxic values of barium and radium taken with food is not clearly established, however Ba toxicity has been reported with ingestions as small as 200 mg [95,96].

  1. McNeill, I.R.; Isoardi, K.Z. Barium Poisoning: An Uncommon Cause of Severe Hypokalemia. Toxicology Communications 2019, 3, 88–90, doi:10.1080/24734306.2019.1691340.
  2. Gustafson PF, Stehney AF. 1985. Exposure Data for Radium Patients. In: Environmental Research Division Annual Report. Argonne, IL: Argonne National Laboratory. ANL-84-103 Part 11:98-180.

The Reviewer's comment: Line 223: In this line you are mentioning HCT116 two times. Please check if this is or not a mistake.

The authors' answer: All changes have been made in the manuscript and highlighted in red.

Tsai et al. observed that selenium-enriched broccoli extract inhibits the growth of HCT116 and HCT116+Chr.3 human colon cancer cells.

The Reviewer's comment: Please uniform trough the text the symbol for micro. Sometimes you use u, and sometimes μ.

The authors' answer: All changes have been made in the manuscript and highlighted in red.

Reviewer 2 Report

Review report:

Title: Natural sources of selenium as functional food products for 2 chemoprevention

Reference. ID: foods-2212856

-The review presents a summary of studies on the anti-cancer activity of the most used functional products rich in Se on the European market. The main dietary sources of selenium compounds and their metabolism were reported. The MS content is interesting and useful for readers.

-Some precisions related to the methodology and improvement should be performed.

-Fig 1. Should be improved, see comments in the pdf file.

-The total number of used articles for the survey was reported to be 27, this number include 11 articles and 24 articles selected at different scale.

Besides references list contains 138 references, so it is not clear if the present review used only 27 selected articles among the total of 138 refs (references list 138). If 27 refs were used among 138 for all MS how the 27 references were specifically used?

-Patents websites (many of them are free) are also a good sources of scientific results with a great potential of commercialization. Addition of patents in my opinion is important, some refs are below:

-¨Patent 1: US20110262564A1

Treatment of Cancer with Selenium Nanoparticles US20110262564A1. Abstract

Novel chemopreventive and chemotherapeutic cancer treatment method using elemental selenium nanoparticles. Cancer cells, especially androgen dependent prostate cancers are exposed to selenium from elemental selenium nanoparticle treatment, and apoptosis is induced in the cancer cells.

Patent 2 US8932649B2

Methods for treating a neoplastic disease in a subject using inorganic selenium-containing compounds. The invention features methods and selenium-containing compositions for treating a neoplastic disease in a subject. In particular, the invention features methods for enhancing sensitivity of a tumor to cancer therapy by treating the tumor with an inorganic selenium-containing (Se) compound and with a cancer therapy, particularly a cancer therapy that also affects the cellular redox status of a tumor cell (e.g., radiation).

-The bioavailability/bioaccesibility of Se need to presented in an independent paragraph. Some indications are reported throughout the MS but this point should be discussed in a section apart.

Below one reference:

-In Vivo Bioavailability of Selenium in Selenium-Enriched Streptococcus thermophilus and Enterococcus faecium in CD IGS Rats.

Gabriela Krausova,1,* Antonin Kana,2 Marek Vecka,3 Ivana Hyrslova,1 Barbora Stankova,3 Vera Kantorova,2 Iva Mrvikova,1 Martina Huttl,4 and Hana Malinska4 Ana Sofia Fernandes, Academic. Antioxidants (Basel). 2021 Mar; 10(3): 463.

Other minors other remarks are listed below and are mentioned in pdf file:

-The quality of all table should be improved:

-Line 28 more recent ref should be provided for example: Int J Mol Sci. 2022 Feb; 23(4): 2215.

Published online 2022 Feb 17. doi: 10.3390/ijms23042215 Potential Role of Selenium in the Treatment of Cancer and Viral Infections. Aseel O. Rataan,1 Sean M. Geary,1 Yousef Zakharia,2,3 Youcef M. Rustum,4,5,* and Aliasger K. Salem1,3,Chiara Riganti, Academic Editor and Marialessandra Contino, 

-The quality, the content and the title of figure should be revised, please see pdf

-the section 3.1. Functional food definition and legislation, is needed in a such article review, its I better to change it by a section dealing with The bioavailability/bioaccesibility of Se

-the number of the section 3.1. should be revised, there isn’t a section 3.2.???

-line 169 and throughout the MS when the ref is cited as: Ip et al. , add the year and the ref as [X] just after the ref and not at the end of the sentence.

Author Response

Response to the comments made by the reviewers

Manuscript ID: foods-2267164

Title: Natural sources of selenium as functional food products for chemoprevention

We would like to thank the Reviewers for their careful review of our manuscript and for providing us with some suggestions to improve its quality. We have carried out a major revision of the manuscript, and we believe the paper has improved significantly.

According to the Reviewers' suggestion, the manuscript has been carefully checked and corrected. The changes in the manuscript have been highlighted in red.

Below we sequentially address all of the points raised by the Reviewers.

Reviewer 2:

Firstly, we would like to express our profound thanks to the Reviewer for devoting time to reviewing our manuscript, the corrections and suggestions. We have carried out a major revision of the manuscript, and we believe the paper has improved significantly.

The Reviewer's comment: Fig 1. Should be improved, see comments in the pdf file.

-The total number of used articles for the survey was reported to be 27, this number include 11 articles and 24 articles selected at different scale.

- Besides references list contains 138 references, so it is not clear if the present review used only 27 selected articles among the total of 138 refs (references list 138). If 27 refs were used among 138 for all MS how the 27 references were specifically used?

The authors' answer: According to the Reviewer's suggestion, we have been corrected the Figure 1.

Analysis concerned on 28 original articles, which met the criteria of the study. Other references are a source of additional information, e.g. cancer mortality, composition and properties of functional foods discussed in the manuscript.

Figure 1. Search and selection methodology of articles used in this review.

The Reviewer's comment: Patents websites (many of them are free) are also a good sources of scientific results with a great potential of commercialization. Addition of patents in my opinion is important, some refs are below:

Patent 1: US20110262564A1

Treatment of Cancer with Selenium Nanoparticles US20110262564A1. Abstract

Novel chemopreventive and chemotherapeutic cancer treatment method using elemental selenium nanoparticles. Cancer cells, especially androgen dependent prostate cancers are exposed to selenium from elemental selenium nanoparticle treatment, and apoptosis is induced in the cancer cells.

Patent 2 US8932649B2

Methods for treating a neoplastic disease in a subject using inorganic selenium-containing compounds. The invention features methods and selenium-containing compositions for treating a neoplastic disease in a subject. In particular, the invention features methods for enhancing sensitivity of a tumor to cancer therapy by treating the tumor with an inorganic selenium-containing (Se) compound and with a cancer therapy, particularly a cancer therapy that also affects the cellular redox status of a tumor cell (e.g., radiation).

The authors' answer: Thank you for indicating patents as valuable sources of scientific results. According to the Reviewer's suggestion, we have been supplemented the manuscript with information contained in patents concerning on Se anti-cancer properties.

The Se anti-cancer properties are not fully understood but may be influenced by antioxidant protection, enhanced immune surveillance, modulation of cell proliferation, inhibition of angiogenesis and tumour cell invasion [56,57]

It has been shown that for MeSeCys, the production of a monomethylated Se metabolite from MeSeCys via β-lyase is a key step in its cancer chemoprevention [126].

  1. Zeng, H.; Combs, G.F. Selenium as an Anticancer Nutrient: Roles in Cell Proliferation and Tumor Cell Invasion. The Journal of Nutritional Biochemistry 2008, 19, 1–7, doi:10.1016/j.jnutbio.2007.02.005.
  2. Knox, S.J., Husbeck, B. Methods for Treating a Neoplastic Disease in a Subject Using Inorganic Selenium-Containing Compounds. Patent Appl. Publ. within the TVPP - United States. PAT: US8932649B2, 13 Jan 2015. Https://Patents.Google.Com/Patent/US8932649B2/En.
  3. Gao, X., Kong, L. Treatment of Cancer with Selenium Nanoparticles, Patent Appl. Publ. within the TVPP - United States. PAT: US2011262564, 18 Jun 2010. Https://Europepmc.Org/Article/PAT/US2011262564#abstract.

The Reviewer's comment: The bioavailability/bioaccesibility of Se need to presented in an independent paragraph. Some indications are reported throughout the MS but this point should be discussed in a section apart.

Below one reference:

-In Vivo Bioavailability of Selenium in Selenium-Enriched Streptococcus thermophilus and Enterococcus faecium in CD IGS Rats.

Gabriela Krausova,1,* Antonin Kana,2 Marek Vecka,3 Ivana Hyrslova,1 Barbora Stankova,3 Vera Kantorova,2 Iva Mrvikova,1 Martina Huttl,4 and Hana Malinska4 Ana Sofia Fernandes, Academic. Antioxidants (Basel). 2021 Mar; 10(3): 463.

The authors' answer: According to the Reviewer's suggestion, we have been added independent paragraph about bioavailability of Se.

2.1. Selenium bioavailability

Se bioavailability depends on the source and chemical forms of the element [61]. Se content in foods determined by combination of geologic, environmental factors and Se supplementation of fertilizers and animal feedstuffs [33]. Chemical forms of Se differ in absorption and conversion into a biochemically active form. Organic sources are assimilated more efficiently and are considered to be less toxic than inorganic compounds [27].

The bioavailability of selenium may also be affected by some other components of the food matrix [62]. Some studies have shown that vitamin E can increase the bioavailability of selenium, while heavy metals and fibre decrease it [63–65]. Dietary sulphur (especially from methionine) may competes with Se for absorption. Additionally, on bioavailability may influence parameters related to the human body i.e. , age, sex, Se status or lifestyle [66].

Current researches are concerned on improving bioavailability and identification of biomarkers of exposure and anti-cancer activity. Interestingly, in recent years studies showed that some Se enrichment of yeasts and lactic acid bacteria represent sources of the more bioavailable organic and less toxic forms of Se [67].

  1. Shini, S.; Sultan, A.; Bryden, W. Selenium Biochemistry and Bioavailability: Implications for Animal Agriculture. Agriculture 2015, 5, 1277–1288, doi:10.3390/agriculture5041277.
  2. Thiry, C.; Ruttens, A.; De Temmerman, L.; Schneider, Y.-J.; Pussemier, L. Current Knowledge in Species-Related Bioavailability of Selenium in Food. Food Chemistry 2012, 130, 767–784, doi:10.1016/j.foodchem.2011.07.102.
  3. Ralston, N.V.C.; Raymond, L.J. Dietary Selenium’s Protective Effects against Methylmercury Toxicity. Toxicology 2010, 278, 112–123, doi:10.1016/j.tox.2010.06.004.
  4. Reeves, P.G.; Leary, P.D.; Gregoire, B.R.; Finley, J.W.; Lindlauf, J.E.; Johnson, L.K. Selenium Bioavailability from Buckwheat Bran in Rats Fed a Modified AIN-93G Torula Yeast–Based Diet. The Journal of Nutrition 2005, 135, 2627–2633, doi:10.1093/jn/135.11.2627.
  5. Vagni, S.; Saccone, F.; Pinotti, L.; Baldi, A. Vitamin E Bioavailability: Past and Present Insights. FNS 2011, 02, 1088–1096, doi:10.4236/fns.2011.210146.
  6. Thomson, C.D. Assessment of Requirements for Selenium and Adequacy of Selenium Status: A Review. Eur J Clin Nutr 2004, 58, 391–402, doi:10.1038/sj.ejcn.1601800.
  7. Krausova, G.; Kana, A.; Vecka, M.; Hyrslova, I.; Stankova, B.; Kantorova, V.; Mrvikova, I.; Huttl, M.; Malinska, H. In Vivo Bioavailability of Selenium in Selenium-Enriched Streptococcus Thermophilus and Enterococcus Faecium in CD IGS Rats. Antioxidants 2021, 10, 463, doi:10.3390/antiox10030463.

Other minors other remarks are listed below and are mentioned in pdf file:

The Reviewer's comment: The quality of all table should be improved:

The authors' answer: According to the Reviewer's suggestion, we have been improved the all table.

The Reviewer's comment: Line 28 more recent ref should be provided for example: Int J Mol Sci. 2022 Feb; 23(4): 2215.

Published online 2022 Feb 17. doi: 10.3390/ijms23042215 Potential Role of Selenium in the Treatment of Cancer and Viral Infections. Aseel O. Rataan,1 Sean M. Geary,1 Yousef Zakharia,2,3 Youcef M. Rustum,4,5,* and Aliasger K. Salem1,3,Chiara Riganti, Academic Editor and Marialessandra Contino, 

The authors' answer: According to the Reviewer's suggestion, we have been improved the references.

Cancer is one of the leading causes of mortality worldwide, accounting for nearly 10 million deaths in 2020. The most common cancer is female breast cancer (11.7%), followed by lung (11.4%), colorectal (10.0%), prostate (7.3%), and stomach (5.6%) cancers [1,2]. According to Global Cancer Statistics (GLOBOCAN 2020: Estimates of Incidence and Mortality Worldwide for 36 Cancers in 185 Countries), the number of people with cancer will continue to grow and reach 28.4 million by 2040 [3]. The main causes of the increase in cancer incidence are environmental and lifestyle factors, with an estimated 30-40% of all cancers preventable through lifestyle and diet [4–6].

  1. Ferlay, J.; Colombet, M.; Soerjomataram, I.; Parkin, D.M.; Piñeros, M.; Znaor, A.; Bray, F. Cancer Statistics for the Year 2020: An Overview. Int. J. Cancer 2021, 149, 778–789, doi:10.1002/ijc.33588.
  2. Siegel, R.L.; Miller, K.D.; Fuchs, H.E.; Jemal, A. Cancer Statistics, 2022. CA A Cancer J Clinicians 2022, 72, 7–33, doi:10.3322/caac.21708.
  3. Sung, H.; Ferlay, J.; Siegel, R.L.; Laversanne, M.; Soerjomataram, I.; Jemal, A.; Bray, F. Global Cancer Statistics 2020: GLOBOCAN Estimates of Incidence and Mortality Worldwide for 36 Cancers in 185 Countries. CA A Cancer J Clin 2021, 71, 209–249, doi:10.3322/caac.21660.
  4. Glade, M.J. Food, Nutrition, and the Prevention of Cancer: A Global Perspective. American Institute for Cancer Research/World Cancer Research Fund, American Institute for Cancer Research, 1997. Nutrition 1999, 15, 523–526, doi:10.1016/s0899-9007(99)00021-0.
  5. Donaldson, M.S. Nutrition and Cancer: A Review of the Evidence for an Anti-Cancer Diet. Nutr J 2004, 3, 19, doi:10.1186/1475-2891-3-19.
  6. Rataan, A.O.; Geary, S.M.; Zakharia, Y.; Rustum, Y.M.; Salem, A.K. Potential Role of Selenium in the Treatment of Cancer and Viral Infections. IJMS 2022, 23, 2215, doi:10.3390/ijms23042215.

The Reviewer's comment: The quality, the content and the title of figure should be revised, please see pdf

The authors' answer: We have been corrected it.

Figure 1. Search and selection methodology of articles used in this review

The Reviewer's comment: The section 3.1. Functional food definition and legislation, is needed in a such article review, its I better to change it by a section dealing with The bioavailability/bioaccesibility of Se.

The authors' answer: According to the Reviewer's suggestion, we have been added information about bioavailability of selenium.  However, according to the comments of the second reviewer, the information on the definition of functional foods was kept and it was improved with missing information. All changes have been made in the manuscript and highlighted in red.

  1. Functional food rich in selenium

According to the European Food Safety Authority (EFSA), “a functional food is defined as a food, which beneficially affects one or more target functions in the body, beyond adequate nutritional effects, in a way that is relevant to either an improved state of health and well-being and/or reduction of risk of disease. A functional food can be a natural food or a food to which a component has been added or removed by technological or biotechnological means, and it must demonstrate their effects in amounts that can normally be expected to be consumed in the diet” [68].

The Functional Food Center (FFC) defines functional food as a natural or processed food that contains known or unknown biologically active compounds, which in defined, effective, non-toxic amounts provide clinically proven and documented health benefits in the prevention and treatment of chronic diseases [69,70].

In United States functional foods are regulated in the same way as conventional foods and dietary supplements. The primary distinction between functional food and food in general is in the claims made for benefits, other than nutritional, attributed to the functional food [71].

The Reviewer's comment: The number of the section 3.1. should be revised, there isn’t a section 3.2.???

The authors' answer: According to the Reviewer's suggestion, we have been corrected number of sections it.

The Reviewer's comment: Line 169 and throughout the MS when the ref is cited as: Ip et al. , add the year and the ref as [X] just after the ref and not at the end of the sentence.

The authors' answer: All changes have been made in the manuscript and highlighted in red.

Other comments marked in the text of the manuscript have been included and highlighted in red.

Round 2

Reviewer 2 Report

Response to the comments made by the reviewers

Manuscript ID: foods-2267164

Title: Natural sources of selenium as functional food products for chemoprevention

The authors respond appropriately all comments.

Some added sentences (in revised paper) need to be revised for English and/or precision to add as mentioned below:

-Line 127 correct the sentence

Se content in foods is determined by combination of geologic, environmental factors and Se supplementation of fertilizers and animal feedstuffs

-line 139 : correct the sentence

Interestingly, in recent years studies have shown (showed) that some Se enrichment of yeasts and lactic acid bacteria represent sources of (the) 140 more bioavailable organic and less toxic forms of Se [67].

-line 147: correct the sentence

 A functional food can be a natural food or a food to which a component has been added or removed by technological or biotechnological means, and it must demonstrate its (their) effects in amounts that can normally be expected to be consumed in the diet [68].

-line 2017: sentence need précision

It is worth noting the toxic values of barium and radium taken with food is not clearly established, however Ba toxicity has been reported with ingestions as small as 200 mg of what???

Author Response

Response to the comments made by the reviewers

Manuscript ID: foods-2267164

Title: Natural sources of selenium as functional food products for chemoprevention

Dear Reviewer, we appreciate all your insightful comments. Thank you for your suggestions.

All changes have been made in the manuscript and highlighted in red.
